# Transcriptomic Profiling Reveals Regulatory Pathways of Tomato in Resistance to Verticillium Wilt Triggered by VdR3e

**DOI:** 10.3390/plants14081243

**Published:** 2025-04-19

**Authors:** Xiao Wang, Qian Tan, Xiyue Bao, Xinyue Gong, Lingmin Zhao, Jieyin Chen, Lei Liu, Ran Li

**Affiliations:** 1State Key Laboratory for Biology of Plant Diseases and Insect Pests, Institute of Plant Protection, Chinese Academy of Agricultural Sciences, Beijing 100193, China; 18832711444@163.com (X.W.); tanqian0201@163.com (Q.T.); baoxiyue3@163.com (X.B.); gxy15670260237@163.com (X.G.); lingminzhao@163.com (L.Z.); chenjieyin@caas.cn (J.C.); 2Western Agricultural Research Center, Chinese Academy of Agricultural Sciences, Changji 831100, China; 3State Key Laboratory of Vegetable Biobreeding, Institute of Vegetables and Flowers, Chinese Academy of Agricultural Sciences, Beijing 100081, China

**Keywords:** tomato, Verticillium wilt, *Verticillium dahliae* race 3, disease resistance, transcriptome sequencing, immune response

## Abstract

Tomatoes are important horticultural crops worldwide. Verticillium wilt is a disease caused by *Verticillium dahliae* that causes serious tomato yield losses. *V. dahliae* can be classified into three distinct races in tomatoes. We identified the specific *VdR3e* gene of *V. dahliae* race 3 and found that VdR3e triggered immune responses in the resistant tomato cultivar IVF6384. We confirmed that VdR3e triggers immune responses in the parents of IVF6384 plants and conducted transcriptome sequencing between male and female IVF6384 plants after VdR3e infiltration to analyze the potential regulatory network response to VdR3e. We found that both parents had a series of detoxification and stress resistance responses to VdR3e, but those of the male IVF6384 parent were concentrated in disease resistance-related signaling pathways. Moreover, several vital differentially expressed genes involved in functional annotation related to plant–pathogen interactions and plant hormone signaling stimulated immune responses in *Nicotiana benthamiana*. This study provides a new and comprehensive perspective on tomato resistance to Verticillium wilt.

## 1. Introduction

Breeding for disease resistance is the foundation of modern agriculture, and model plants in genetics and genomics provide resources for molecular biology exploration. Extracellular and intracellular receptors translate the recognition of molecular patterns encoded by extracellular microorganisms or virulence effectors delivered by intracellular pathogens into defense responses [1]. Plants have evolved two layers of innate immunity to detect and respond to biological attacks. The first layer comprises pattern recognition receptors (PRRs), located on the surface of plant cells, including receptor-like kinases (RLKs) and receptor-like proteins (RLPs), which recognize conserved pathogen molecular patterns (PAMPs) and activate pattern-triggered immunity (PTI) [2]. Flagellin is the most common PAMP found in plants. Flagellin receptor 2 (FLS2) in *Arabidopsis thaliana* is an RLK that recognizes the highly conserved 22-amino-acid epitope flagellin 22 (flg22) present at the N-terminus of flagellin and activates the downstream immune response [3]. The immune receptor complex FLS2-BAK1 and the related cytoplasmic kinases BIK1 and PBL1 phosphorylate the same S-cluster in Ca^2+^/H^+^ exchangers (CAXs) to regulate Ca^2+^ signaling during immunity [4]. Pathogens evade or inhibit PTI by secreting effectors that target or attack the direct downstream processes of PRR signaling and other subsequent events, as host–pathogen interactions continue to evolve [5]. This leads to effector-triggered susceptibility (ETS) that facilitates pathogen proliferation and disease development. Plants then evolved a second layer of immunity, i.e., the nucleotide-binding domain and leucine-rich repeat (NLR) proteins within the cell. NLR proteins sense the effectors that activate effector-triggered immunity (ETI), causing hypersensitive response (HR) and programmed cell death [6]. The NLR proteins RPM1 and RPS2 in *A. thaliana* bind to guard RPM1-interacting protein 4 (RIN4) to form a complex that is inactive in the absence of pathogens. The effector proteins AvrB and AvrRpml are recognized by RIN4 and phosphorylate RIN4 when pathogens infect *A. thaliana*. The resistance proteins RPM1 and RPS2 detect phosphorylated RIN4 and activate the resistance response [7,8]. AvrRpm2 is an effector related to AvrRpm1, with 50% sequence identity. Both have the same biochemical function, relying on the conserved H-Y-D motif (three conserved residues H63, Y122, and D185 in their catalytic domains), and can modify RIN4 by activating RPM1 [9]. PTI and ETI are not completely independent systems: they work together to enhance plant immune responses by sharing downstream immune signaling molecules [10]. A series of downstream signal transduction processes are activated, including bursts of reactive oxygen species (ROS), calcium fluxes, mitogen-activated protein kinase (MAPK) cascades, and plant hormone signal transduction and regulation, upon the recognition of pathogens by pattern recognition receptors on the cell membrane and NLRs within the cell. Moreover, flg22 induces the phosphorylation of *A. thaliana* CDPK5 protein, and CDPK5 phosphorylates the RbohD protein, thereby regulating ROS production and enhancing resistance to the bacterial pathogen *Pseudomonas syringae* pv. *tomato* (*Pst*) DC3000 [11]. The signaling pathways regulated by jasmonic acid (JA) and salicylic acid (SA) have been extensively studied among the plant-hormone-mediated defense mechanisms and are central components of the immune system [12]. The signal from these pathways is further amplified and eventually passed into the cell nucleus, causing the plant to initiate relevant defense responses, such as the reprogramming of defense-related gene transcription and callose deposition on the cell wall. Members of the WRKY transcription factor family are involved in plant immunity. MPK3 and MPK6 phosphorylate the transcription factors WRKY22 and WRKY29 [13], activate the transcription of relevant immune defense-related genes, and positively regulate the PTI response in plants. WRKY22 in *A. thaliana* positively regulates the expression of aminocyclopropane-1-carboxylic acid synthase (*ACS*) and aminocyclopropane-1-carboxylic acid oxidase (*ACO*) genes and promotes ethylene production [14]. However, the activation of the transcriptome factors WRKY25 and WRKY33 inhibits the transcription of genes related to immune defense [15].

Tomatoes are native to South America and cultivated worldwide, are highly valuable as food, and serve as a model plant for studying host disease resistance [16]. China is the largest tomato producer in the world, with its production volume exceeding that of India (68.34 and 20.69 million tons in 2022, respectively), according to the Food and Agriculture Organization [17]. In the cultivation and production of tomatoes, there are many soilborne diseases that negatively affect the yield and quality of tomatoes and cause huge economic losses [18]. Diseases such as Verticillium wilt (VW), Rhizoctonia root rot, southern blight, and so on can cause wilt and death of seedlings and tomato plants [19,20]. Chemical control is currently the main method used for preventing tomato diseases; however, the long-term, large-scale use of pesticides not only destroys the ecological environment but also leads to pathogen resistance [21]. Breeding resistant tomato cultivars is one of the most effective methods of controlling tomato diseases and reducing pathogen resistance to fungicides. The rapid and efficient cultivation of disease-resistant pesticides decreases production costs and improves the agricultural ecological environment. Research on resistance genes is key to breeding disease-resistant cultivars and thus for agricultural production [22]. Many resistance proteins involved in the immune responses in tomatoes have been identified. The direct homolog of FLS2 has been cloned into tomatoes and is activated by flg22 [23,24]. Ethylene-inducing xylanase (EIX) is a potent elicitor of plant defense responses. Xylanase is a fungal PAMP recognized by the tomato RLPs, i.e., LeEIX1 and LeEIX2. Although both LeEIX1 and LeEIX2 can bind to EIX, only LeEIX2 can trigger a signal that initiates a PTI immune response [25]. The *LeEIX* locus may control resistance to necrotrophic fungal pathogens, whereas resistance to biotrophic fungal pathogens is regulated by quantitative trait loci [26]. AvrPto is a small hydrophilic protein delivered to plant cells via the bacterial type III secretion system. Tomato *Pto* encodes a serine/threonine kinase that confers resistance to *Pst* strains expressing AvrPto [27]. After *Pst* infects tomatoes, AvrPto interacts with Pto. Prf belongs to the NLR and recognizes the Avrpto-Pto complex to activate downstream ETI responses [28]. Prf may also act independent of Pto to activate a distinct set of defense-related genes [29]. Nrc2 and Nrc3 are helper NLRs, codependent with Prf/Pto, and activate MAPK signaling to induce immunity in tomatoes [30]. SlPti1 is the first protein that interacts with Pto and belongs to the receptor-like cytoplasmic kinases (RLCKs) family. SlPti1 is phosphorylated by Pto and participates in hypersensitivity reactions [31]. Three ethylene response factors (ERFs) (Pti4/Pti5/Pti6) interact with Pto in the yeast two-hybrid system. These ERFs act downstream of the Pto pathway and promote the expression of pathogenesis-related (PR) genes [32]. SlPti5 positively regulates the tomato immune response to *Botrytis cinerea* through the ROS cleanup system and hormone signaling pathways [33].

Verticillium wilt (VW) is one of the most important diseases of tomato caused by the soilborne fungus *Verticillium dahliae*, which has a wide host range (over 400 cultivated and non-cultivated host species). It is also able to colonize the plant′s vascular system, restricting water and nutrient transport, and ultimately causing wilting and decline of the plant host [20,34,35,36]. Infected tomato leaves form characteristic “V”-shaped necrotic lesions with yellow halos. As *V. dahliae* infection progresses, the entire leaves wilt and drop, the vascular bundles of the infected plant turn brown, and growth is stunted, ultimately affecting yield [37]. The gene-for-gene hypothesis was proposed by Flor in 1942 based on a study of the specificity of flax and races of the flax rust pathogen *Melampsora lini* [38]. Host plants possess resistance (*R*) and susceptibility (*r*) genes, whereas pathogens possess avirulence (*Avr*) and virulence (*vir*) genes. The resistance of plants is controlled by genotype and is usually determined by the incompatible interaction between the *R* gene from the plant and the *Avr* gene from the pathogen. The other three interactions lead to susceptibility [39]. The tomato *Ve1* gene is considered a classic *R* gene that adheres to Mendelian genetics and encodes a receptor protein that recognizes the fungal effector Ave1 in response to *V. dahliae* infection. *V. dahliae* that does not cause disease in tomato cultivars with the *Ve* locus is classified as race 1, whereas those that can cause disease in such cultivars are designated as race 2 or 3 [40,41]. The resistance gene *Ve* locus against *V. dahliae* race 1 originated from *Solanum pimpinellifolium*. The *Ve* locus was found to be located on the short arm of chromosome 9 in tomato using restriction fragment length polymorphism [42,43]. DNA sequencing revealed that the *Ve* locus includes two homologous genes, *Ve1* (3162 bp) and *Ve2* (3420 bp), both of which encode RLPs. Only *Ve1* mediates resistance to *V. dahliae* race 1 in tomato [44]. The results of comparative genomic analyses and genetic experiments showed that *Ave1* is the *Avr* gene of *V. dahliae* race 1. Ave1 in tomatoes is recognized by the R protein encoded by *Ve1* and induces host immune responses [45]. The results of selfing and crossing tests between resistant and susceptible cultivars showed that the segregation (from resistant to susceptible) ratio of F_2_ plants was 3:1. This indicated that resistance to race 2 is controlled by a single dominant gene locus, *V2* [41]. The *Avr* gene *Av2* in race 2 was identified and found to activate *V2*-mediated resistance using comparative genomics [46]. Subsequently, comparative genomic analysis of three races identified a protein, VdR3e, specifically secreted by race 3 [47]. However, the mechanisms of the resistance genes or loci in race 3 have not been explored.

We have previously identified the specific gene *VdR3e* and that VdR3e triggers immune responses in the resistant tomato cultivar IVF6384 [47,48]. In this study, we confirmed that VdR3e triggers an immune response in the parents of IVF6384 tomatoes and conducted transcriptome sequencing between male and female IVF6384 plants after VdR3e infiltration to analyze the regulatory network response to VdR3e. Both parents exhibited a series of responses to VdR3e, but differentially expressed genes (DEGs) in the male parent of IVF6384 are concentrated in disease-resistance-related signaling pathways. Furthermore, several DEGs involved in functional annotation related to plant–pathogen interaction and plant hormone signal transduction stimulate immune responses in *N. benthamiana*. This study provides a new and comprehensive perspective on tomato resistance to VW.

## 2. Results

### 2.1. VdR3e Protein Triggered Immune Response in Parents of Solanum lycopersicum Cultivar IVF6384

VdR3e may be an effector that triggers cell death and various other defense responses [48]. In addition, IVF6384 is a tomato cultivar resistant to *V. dahliae* race 3 strain HoMCLT [47]. The recombinant protein VdR3e expressed in prokaryotes was infiltrated into tomato leaves at concentrations of 50, 100, and 150 mg/mL to confirm that VdR3e induces cell death and other defense responses in female and male parents of IVF6384 tomato plants. The results showed that VdR3e infiltration at all three concentrations at 24 h post infiltration (hpi) caused leaf necrosis in both the female and male parents of IVF6384 plants (Figure 1A). Moreover, the ROS accumulation was substantially higher in leaf tissues after VdR3e infiltration at 24 hpi in both the female and male parents of IVF6384 plants than at baseline (Figure 1B). ROS accumulation in the male parent leaves of IVF6384 plants was higher than that in the female parent leaves of IVF6384 (Figure 1B,C). These results indicate that VdR3e stimulates immune responses in the female and male parents of IVF6384 tomato plants.

### 2.2. Identification of DEGs in Parents of IVF6384 Tomato Plants After VdR3e Protein Infiltration Using RNA-Seq

We hypothesized that a series of biological processes change with changes in DEGs because of the necrotic phenotype observed on the IVF6384 parent leaf tissues after VdR3e protein infiltration. Thus, RNA-seq was performed to explore the genetic resistance network of the IVF6384 parents after VdR3e protein infiltration. The leaf tissues of the female and male parents of IVF6384 tomato plants infiltrated with phosphate-buffered saline (PBS) and recombinant VdR3e protein at 24 hpi and leaves without infiltration (control, CK) were independently collected for sequencing. Six groups of samples were used for sequencing, including CK plants, infiltration of PBS and VdR3e in the female parent, and CK and infiltration of PBS and VdR3e in the male parent. All clean reads were mapped to the tomato genome at an average rate of 96.46% (Appendix A). Thousands of DEGs were identified in the female and male parents after infiltration with PBS and VdR3e (Appendix A). The male parent of IVF6384 plants had twice as many DEGs as the female parent, and the number of DEGs in the VdR3e/CK treatment was higher than that in the PBS/CK treatment (Figure 2A). The Venn diagram in Figure 2B displays the common and specific DEGs between the PBS/CK and VdR3e/CK groups in the female and male parents of IVF6384 plants, respectively. In the male parent, 761 and 1265 genes were specifically expressed in the PBS/CK and VdR3e/CK groups, respectively; 2090 genes were expressed in both the PBS/CK and VdR3e/CK groups (Figure 2B). In the female parent, 450 and 566 genes were specifically expressed in the PBS/CK and VdR3e/CK groups, respectively; 1049 genes were expressed in both the PBS/CK and VdR3e/CK groups (Figure 2B). Additionally, the fold changes for these groups were calculated, and most DEGs had log_2_FoldChange values between 1 and 10, with a few having values exceeding 10 (Figure 2C). Therefore, many genes were differentially expressed in the parent IVF6384 cells in response to VdR3e infiltration. The male parent contained more DEGs than the female parent of IVF6384 plants.

### 2.3. Functional Analysis of DEGs in Parents of IVF6384 Plants in Response to VdR3e Infiltration

GO functional enrichment and KEGG pathway enrichment were analyzed to determine the specific DEGs responses to VdR3e. The DEGs of the male parent VdR3e/CK were enriched in 114 categories, such as the response to wounding and regulation of growth (*p* < 0.05), according to GO annotation (Appendix A). Moreover, the DEGs were enriched in 78 categories in the female parent plants in the VdR3e/CK group, such as secondary metabolic processes (*p* < 0.05) (Appendix A). The GO terms included biological processes, cellular components, and molecular functions. Regarding molecular functions, DEGs were significantly enriched in the proton-transporting ATP synthase activity (GO:0046933) of the male parent plants in the VdR3e/CK groups, whereas the DEGs of the female parent VdR3e/CK plants were enriched in three GO terms (GO:0052716, GO:0016682, and GO:0016679), which were related to oxidoreductase activity (Figure 3A). The enriched DEGs in the male parent VdR3e/CK group were associated with response to wounding or toxic substance (GO:0009611, GO:0098869, GO:1990748, and GO:0097237) and growth regulation (GO:0040008) in the biological processes. The DEGs of female parent VdR3e/CK plants were enriched in the metabolism and catabolism processes of various substances, including phenylpropanoid (GO:0046271 and GO:0009698), lignin (GO:0046274 and GO:0009808), and chlorophyll (GO:0015995 and GO:0015994) (Figure 3A). The DEGs of VdR3e/CK plants in the cellular components were enriched in chloroplasts related to photosynthesis. The DEGs of the male parent VdR3e/CK plants were enriched in the membrane and complex (GO:0042651, GO:0019898, GO:0009654, and GO:0045261), and the DEGs of the female parent VdR3e/CK plants were enriched in photosystem I (GO:0009538 and GO:0009522) (Figure 3A). GO functional enrichment analysis revealed that, compared with PBS/CK, the DEGs of VdR3e/CK were primarily enriched in chloroplasts associated with photosynthesis. The male parent focused on eliminating the toxic effects of ROS on cells, while the female parent regulated the metabolic processes of secondary metabolites to enhance immune responses. These results indicated that both parents developed coping responses to VdR3e stress 24 h after infiltration using different and specific response processes.

The results of KEGG network analysis showed that 143 pathways matched the DEG response to VdR3e infiltration (Appendix A). Several DEGs were highly enriched in the plant hormone signal transduction, plant–pathogen interactions, phenylpropanoid biosynthesis, MAPK signaling, and glutathione metabolism pathways (Figure 3B). The DEGs in the male parent VdR3e/CK plants were considerably more enriched in the plant hormone signal transduction, plant–pathogen interactions, and MAPK signaling pathways than those in the female parent VdR3e/CK plants (Figure 3B). However, the DEGs in the female parent VdR3e/CK group were more enriched than those in the male parent VdR3e/CK group in the biosynthesis of amino acids and glycolysis/gluconeogenesis (Figure 3B). The DEGs of both the male and female parents were enriched in photosynthesis, and the number of DEGs in male parents was higher than that in female parents (Figure 3B). Together, these results showed that VdR3e infiltration stimulated several immune responses in IVF6384 parents, but the regulatory pathways differed between the male and female parents. Compared with the DEGs of the female parent of IVF6384 plants, the DEGs of the male parent are concentrated in disease resistance-related signaling pathways.

### 2.4. Analysis of Typical Resistance-Related Pathways Enriched in DEGs

DEGs are typically enriched in the plant–pathogen interaction pathway among the KEGG resistance-related pathways. Plant–pathogen interaction is directly involved in plant resistance to pathogens and induces a series of signal transductions, stimulating the host defense system of the plant to resist pathogens, resulting in a disease resistance response. The DEGs specifically induced by VdR3e from the female and male parents were enriched at 15 key points in the plant–pathogen interaction pathway, and the relevant information on these DEGs is presented in Figure 4A and Appendix A. In PAMP-triggered immunity, the DEGs induced by VdR3e from both the female and male parents were enriched in calmodulin- and calcium-binding protein CML (CaMCML) and pathogenesis-related protein 1 (PR1); the DEGs of the male parent were specifically enriched in FLS2, WRKY22, EIX1/2, and Pit1; and the DEGs of the female parent were inimitably assigned to Prf (Figure 4A). In effector-triggered immunity, both male and female DEGs were enriched in 3-ketoacyl-CoA synthase (KCS); the DEGs of the male parent were enriched in the suppressor of the G2 allele of SKP1 (SGT1), RPM1, and heat shock protein 90 kDa (HSP90) (Figure 4A). A total of 23 DEGs were enriched in the plant–pathogen interaction pathway; 2 genes involved in CaMCML and Prf were upregulated in the female parent VdR3e/CK plants, and 13 genes were upregulated in the female parent VdR3e/CK plants (Figure 4B).

The DEGs were highly enriched in the plant hormone signal transduction pathway, being most highly enriched in the male parents in the VdR3e/CK group. Plant hormones are conserved in the plant kingdom, regulating different growth and development processes, as well as defense responses to abiotic and biotic stressors. Plant hormones are central regulators of plant immunity and usually do not act alone but rather synergistically, antagonistically, or additively interact with each other or with other signaling molecules [12]. The DEGs induced by VdR3e were enriched in the auxin, ethylene, cytokinin, brassinosteroid, salicylic acid, and abscisic acid signal transduction pathways among the plant hormone signal transduction pathways (Figure 5 and Appendix A). The brassinosteroid signal transduction pathway was only assigned to one female parent DEG, cyclin D3 (CYCD3), which was upregulated (Figure 5A,B). Only two DEGs from the male parent were specifically assigned to abscisic acid signaling: one downregulated gene of the male parent was assigned to the abscisic acid receptor PYR/PYL, and one upregulated gene was assigned to protein phosphatase 2C (PP2C) (Figure 5A,B). A total of 30 DEGs were assigned to other plant hormone signal transduction pathways: 20 genes were significantly differentially expressed in male parent VdR3e/CK plants; 3 genes were significantly differentially expressed in both male and female parent VdR3e/CK plants; and 7 genes were significantly differentially expressed in female parent VdR3e/CK plants.

### 2.5. DEGs Induced by VdR3e Play Important Roles in Immune Response

We selected 10 DEGs for immunity function analysis to investigate whether these specifically expressed genes involved in plant–pathogen interaction and plant hormone signal transduction pathways function in immune induction. Six genes were specifically expressed in the male parent in response to VdR3e (*SlCML*, *SlPTI1*, *SlSGT1*, *SlHSP90A*, *SlHSP90B*, and *SlPYL*), two genes were specifically expressed in the female parent in response to VdR3e (*SlERF1* and *SlCYCD3*), and two genes were upregulated in both parents (*SlPR1* and *SlCALM*) (Appendix A). These genes were cloned into the pCAMBIA-1300 expression vector and transiently transformed into *N. benthamiana* leaves via *Agrobacterium* infiltration. The positive control infiltrated with BAX showed leaf necrosis, whereas the *N. benthamiana* leaves infiltrated with GFP whereas the screening genes did not, five days after *Agrobacterium* infiltration (Figure 6A). The expression of *N. benthamiana* defense response genes was detected 2 days after *Agrobacterium* infiltration to further define the role of these genes in inducing immunity that is not associated with cell death. Nine genes of *N. benthamiana* were selected as markers, including SA marker genes *NbPR1*, *NbPR2*, and *NbPR5*; JA marker genes *NbPR4*, *NbLOX*, and *NbPDF1.2*; ET marker genes *NbERF1* and *NbEIN*; and HR marker gene *NbHSR203*. The nine DEGs induced the expression of defense response marker genes after transient expression in *N. benthamiana* leaves (*SlCML*, *SlPTI1*, *SlSGT1*, *SlHSP90A*, *SlHSP90B*, *SlPYL*, *SlERF1*, *SlCYCD3*, and *SlPR1*), whereas *SlCALM* downregulated the expression of defense response marker genes after transient expression in *N. benthamiana* leaves (Figure 6B). The DEGs specifically expressed in male parents broadly upregulated the expression of defense response marker genes, whereas the DEGs specifically expressed in female parents upregulated the expressions of *NbLOX* and *NbPDF1.2* (Figure 6B). These results overall indicate that resistance-related genes that are differentially expressed in IFV6384 parents induce an immune response.

## 3. Discussion

Tomatoes are an important horticultural crop grown worldwide, and VW is a major biological constraint in temperate regions [34]. Tomato VW is caused by *V. dahliae*, a soil-borne vascular disease with a wide range of hosts that can cause serious harm to tomatoes [49]. Tomatoes can be divided into three races based on host resistance to *V. dahlia* [41]. We have previously identified a protein specifically secreted by race 3, VdR3e, that plays vital roles in virulence and induces plant immune responses [47,48]. In addition, the tomato cultivar IVF6384 is resistant to race 3 [47,48]. However, the mechanism underlying the resistance of IVF6384 to race 3 had not yet been explored. Thus, we explored the gene regulatory network underlying the molecular mechanism of VdR3e in the pathogenic differentiation in the parents of IVF6384 tomato plants. The VdR3e protein stimulates immune responses in both the female and male parents of IVF6384 plants, and thousands of genes were differentially expressed in the parents of IVF6384 after VdR3e infiltration (Figure 1 and Figure 2). The DEGs were significantly enriched in several typical resistance-related pathways induced by VdR3e, and some of these DEGs played important roles in the immune response (Figure 3, Figure 4, Figure 5 and Figure 6). The regulatory pathways of tomato resistance to VW triggered by VdR3e were preliminarily analyzed.

The leaf necrosis phenotype after VdR3e protein infiltration suggested a series of defensive responses to VdR3e in the IVF6384 parents. RNA-seq analysis was performed to explore the similarities and differences between the IVF6384 parents in response to VdR3e as well as the source of resistance to IVF6384. Many genes were differentially expressed in the parental IVF6384 cells in response to VdR3e infiltration, with the IVF6384 male plants containing more DEGs than the IVF6384 female plants (Figure 2). Chloroplasts are direct or indirect targets for various pathogens, and the expression of genes regulating photosynthesis is affected when there is a pathogen infection [50]. The enrichment of DEGs in chloroplasts for VdR3e/CK confirmed that VdR3e indirectly activates the parental immune response. After VdR3e protein infiltration at 24 hpi, male-specific DEGs were significantly enriched in GO terms related to response to wounding, cellular detoxification, and growth regulation. Following the combined infiltration of VdR3e protein, ROS accumulation was observed to be higher in male parent leaves than in female parent leaves of IVF6384 (Figure 1 and Figure 3A). It is hypothesized that the male parent is induced by the VdR3e protein to produce a large amount of ROS, thereby activating relevant pathways for wounding response and cellular detoxification to reduce or eliminate toxic superoxide radicals or hydrogen peroxide. In contrast, female-specific DEGs induced by VdR3e were enriched in phenylpropanoid, lignin, and chlorophyll metabolism (Figure 3A). Phenylpropanoid metabolism plays a critical role in plant development and survival, producing a variety of metabolites such as flavonoids and lignin [51]. Lignin, as the primary structural component of the secondary cell wall in vascular plants, acts as a physical barrier against pathogen invasion, and its composition and content influence disease severity in plants [52]. It is hypothesized that the female parent regulated the metabolic processes of secondary metabolites to enhance immune responses. KEGG pathway analysis demonstrated that male-specific DEGs were mainly enriched in plant hormone signaling, plant–pathogen interaction, and the MAPK signaling pathway. The enrichment pathway of female-specific DEGs is less than male-specific DEGs (Figure 3B). In the plant–pathogen interaction, the male parent of IVF6384 exhibited significant enrichment in key resistance sites such as CaMCML, PR1, FLS2, WRKY22, EIX1/2, Pit1, SGT1, RPM1, and HSP90, with most DEGs showing upregulated expression (Figure 4). Among the plant hormone signal transduction, DEGs of the male parent were most enriched here, primarily in auxin, ethylene, cytokinin, salicylic acid, and abscisic acid signaling pathways. The expression of most auxin- and cytokinin-related genes was downregulated, while the expression of ERF in the ET signaling pathway and PR1 in the SA signaling pathway was upregulated, both of which play crucial roles in plant immunity. In the abscisic acid signaling pathway, PP2C expression was upregulated (Figure 5). As an important class of protein phosphatases, PP2C negatively regulates ABA signaling, participates in plant disease resistance signaling pathways, and plays a regulatory role in intracellular redox balance [53] (Figure 3B). Overall, these results indicate that both parents have a series of resistance responses to VdR3e and that the DEGs of the male parents of IVF6384 plants are concentrated in disease resistance-related signaling pathways.

Plants have evolved two layers of innate immune systems to defend against pathogen invasion. The pathogen proteins secreted in plant cells trigger PTI and ETI [1,2]. *Ave1* induces the expression of several resistance genes, including *PR1*, *PR2*, and peroxidase-related genes [54]. VdR3e is a virulence factor that can interact with BAK1 as a PAMP to trigger immune responses, activating the expressions of *PR1*, *PR2*, and several HR and JA marker genes [48]. We found many genes that were significantly differentially expressed in the IVF6384 parents after VdR3e infiltration. Ten DEGs typically involved in plant–pathogen interaction and plant hormone signal transduction were significantly enriched and selected for immunity analysis (Figure 4 and Figure 5). Among the ten genes, except for *SlCML* and *SlCALM*, which are related to the calcium ion signaling pathway, the expression of the *NbPDF1.2* gene after transient expression of the other eight genes was significantly upregulated. It indicated that these genes in response to VdR3e may be closely related to the JA pathway. The marker genes of the SA, JA, and ET pathways were upregulated after transient expression of *SlHSP90A*, *SlPYL*, *SlERF*, and *SlPR1*, suggesting that these genes are simultaneously involved in multiple defense pathways. *NbHSR203* upregulation was significant after the transient expression of *SlHSP90A*, *SlERF*, and *SIPYL*, suggesting that these genes are associated with programmed cell death (Figure 6). HSPs are a class of stress proteins that have protective effects and play important roles in protein folding and unfolding as well as the regulation of cellular immune responses [55]. *SlPYL* is a major gene involved in regulating tomato fruit development. The expression of most *SlPYL* genes is downregulated in tomato leaves under dehydration stress [56]. ERFs are a family of transcription factors that play an important role in plant immunity. SlERF2 plays a key role in the SA, JA, and ROS signal transduction pathways and enhances tomato resistance against *Stemphylium lycopersici* via directly or indirectly regulating the expressions of *Pto*, *PR1b1*, and *PR-P2* [57]. Our conclusions are similar to those of the above studies. These findings not only confirm that the selected genes have immune-inducing functions but also identify their specific pathways of action and pathway specificity in *N. benthamiana* defense mechanisms. This study provides important information for understanding the molecular mechanisms underlying the plant immune response to VdR3e and contributes genetic resources for the subsequent breeding of crop varieties with increased stress resistance. However, the immune mechanism of these DEGs needs to be elucidated in tomatoes (IVF6384 plants and the parents of IVF6384 plants), and the immune mechanism of tomatoes (IVF6384 plants and the parents of IVF6384 plants) requires further research.

## 4. Materials and Methods

### 4.1. Plant Growth Conditions

The *Solanum lycopersicum* cultivar IVF6384 male (Zhongfan No. 144) and female parents (Zhongfan No. 310) were grown and maintained in a greenhouse at 28 °C with a light cycle of 16 h light and 8 h dark.

### 4.2. Infiltration of VdR3e Protein

Protein VdR3e at concentrations of 50 μg/mL, 100 μg/mL, and 150 μg/mL was employed for infiltration experiments with PBS as a control. Using a 1 mL syringe, the protein was infiltrated into the leaves of 5-week-old tomato plants. Each experiment involved infiltrating three leaves from three different plants, and the procedure was repeated at least three times.

### 4.3. ROS Activity

ROS accumulation was visualized using 3,3′-diaminobenzidine (DAB) solution (Aladdin, Shanghai, China). Tomato leaves were examined 24 h after infiltration of 5 μg/mL VdR3e protein and PBS buffer. The infiltrated leaves were collected, rinsed with ddH_2_O, and immersed in a prepared DAB solution (0.5 g DAB dissolved in 500 mL ddH_2_O, pH adjusted to 3.8) [58]. The samples were then incubated on an orbital shaker in the dark for 8 h. Subsequently, the leaves were washed with ddH_2_O to remove excess DAB from the surface and decolorized using 95% ethanol. After complete removal of chlorophyll, the leaves were examined under a stereoscopic microscope. All experiments were repeated three times.

### 4.4. RNA Extraction, Library Construction, and RNA Sequencing Analysis

Leaves from 5-week-old IVF6384 parents were infiltrated with 5 μg/mL VdR3e protein and PBS buffer as a control. Samples were collected 24 h post-infiltration. All samples were promptly frozen in liquid nitrogen and stored at −80 °C until needed. Total RNA was extracted using the FastPure Universal Plant Total RNA Isolation Kit (Vazyme, Nanjing, China) and prepared for sequencing, with three biological replicates for each sample. Genomic DNA was removed by DNase treatment (Aidlab, Beijing, China). There were 6 groups that male parents infiltrated with 5 μg/mL VdR3e protein, infiltrated with PBS buffer and no treatment (CK), and female parents infiltrated with 5 μg/mL VdR3e protein, infiltrated with PBS buffer and no treatment (CK). Moreover, 3 replicates were in each group, with a total of 18 RNA samples. The RNA samples were sent to the transcriptome for sequencing, and the returned sequencing data were collated. Clean reads were obtained by using the Illumina HiSeq X-Ten to remove the adapter and low-quality reads (quality score > Q20). Clean reads were mapped to tomato reference genome sequences using Hierarchical Indexing for Spliced Alignment of Transcripts (HISAT, v0.1.6-beta) data comparison software [59]. The data presented in this article have been deposited in the National Center for Biotechnology Information (NCBI) Sequence Read Archive (http://www.ncbi.nlm.nih.gov/sra/, accessed on 21 February 2025); the accession number is PRJNA1205222.

### 4.5. Identification of Differentially Expressed Genes (DEGs)

Fragments Per Kilobase of the transcript, per million mapped reads (FPKM), was used to determine expression values. DeSeq2 R package (1.46.0) was used to calculate the FPKM of genes in each sample [60]. The fold change in gene expression value was calculated by FPKM treat/FPKM control. Transcripts were identified as DEGs between treatment and control with parameters of fold change > 2 and *p*-value < 0.05.

### 4.6. Functional Annotation

The DEGs were evaluated for their biological functions using Gene Ontology (GO) analysis via the TBtools-II (Toolbox for Biologists) v2.119 [61] and KEGG [62] analysis through the KEGG Orthology (KO)-Based Annotation System (KOBAS). GO analysis covered three categories: cellular component, molecular function, and biological process. Significance was assessed using the Pearson chi-square test (*p* < 0.05) provided by the TBtools-II tool. The dotplot was plotted by https://www.bioinformatics.com.cn (last accessed on 10 October 2024), an online platform for data analysis and visualization.

### 4.7. Agrobacterium Infiltration Assays

The corresponding gene fragments were amplified from the cDNA of the IVF6384 parents. Then, these fragments were cloned into the pCAMBIA-1300 vector and introduced into the *Agrobacterium* strain GV3101. *Agrobacterium* clones transformed with the expression plasmid were picked and inoculated into 1 mL of LB liquid medium supplemented with Kana and Rif and cultured at 28 °C on a shaker at 200 rpm for 12 h. Furthermore, 50 μL of the overnight culture was inoculated into 5 mL of LB liquid medium supplemented with Kana and Rif for further expansion culture at 28 °C and 200 rpm for another 12 h. The bacteria were washed by centrifugation at 4000 rpm for 8 min at room temperature and resuspended in a solution of 10 mM MES, 10 mM MgCl_2_, and 0.2 mM acetosyringone (pH = 6), adjusted to OD_600_ = 1.0, and then kept in the dark at 28 °C for 3 h. The *Agrobacterium* suspension was infiltrated into the leaves of *N. benthamiana* using a 1 mL syringe. Each experiment was conducted on three leaves from three individual plants and repeated at least three times.

### 4.8. Reverse Transcription and Quantitative PCR (RT-qPCR)

Leaves from tomato plants infiltrated with *Agrobacterium* were collected for RNA extraction. cDNA synthesis was performed using 2 µg of RNA per sample with the TranScript One-Step gDNA Removal and cDNA Synthesis SuperMix Kit (Trans, Beijing, China). Quantitative PCR (qPCR) was carried out using the 2×RealStar Fast SYBR qPCR Mix (Low ROX) (GenStar, Beijing, China). Relative quantification of RT-qPCR data was measured using the 2^−∆∆Ct^ analysis method, with normalization to the tomato gene *Slβ-actin* for mRNA expression levels. Each experiment was conducted with three biological replicates, and each replicate was performed in triplicate technical replicates. Specific primers used are listed in Appendix A.

### 4.9. Statistical Analysis

Gene expression data were analyzed using GraphPad Prism v8.3.0. A parametric Student’s *t*-test was used to determine whether the expression of the marker gene was statistically different from GFP and DEG after transient transformation of tobacco (* for *p* < 0.05, ** for *p* < 0.01, *** for *p* < 0.001 and **** for *p* < 0.0001).

## 5. Conclusions

We confirmed that the VdR3e protein triggers an immune response in the parents of IVF6384 tomato plants and conducted transcriptomic profiling to identify the regulatory pathways of male and female parents of IVF6384 in defense after VdR3e infiltration. Functional annotation indicated the DEGs are related to immune responses such as plant–pathogen interactions and plant hormone signaling. Both male and female parents exhibited a series of detoxification and stress resistance responses to VdR3e, but the DEGs in the male parent of IVF6384 plants concentrated in disease resistance-related signaling pathways. Moreover, several typical DEGs involved in functional annotation related to plant–pathogen interaction and plant hormone signal transduction stimulated immune responses in *N. benthamiana*. This study provides a new and comprehensive perspective on tomato resistance to VW.

## Figures and Tables

**Figure 1 plants-14-01243-f001:**
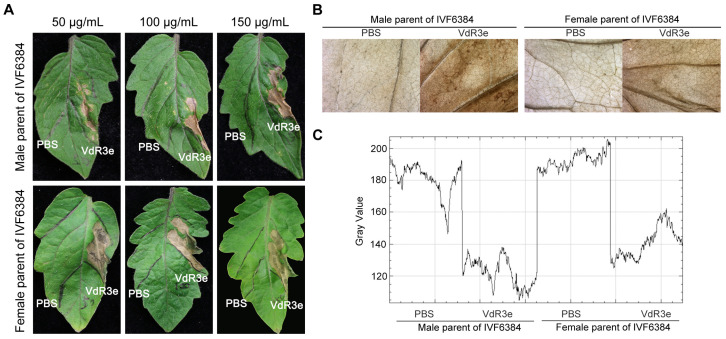
Analysis of the immune response induced by VdR3e protein in IVF6384 parents. (**A**) Necrotic phenotype caused by infiltrating VdR3e protein into IVF6384 parent leaves at 24 h. (**B**) Detection of ROS accumulation induced by VdR3e. (**C**) ROS accumulation levels in tomato leaves were measured after the transient expression of VdR3e protein and the PBS control for 24 h. ImageJ (Java 1.8.0 322 64-bit) was used to analyze the gray value. Values represent the means ± the SE of three independent samples.

**Figure 2 plants-14-01243-f002:**
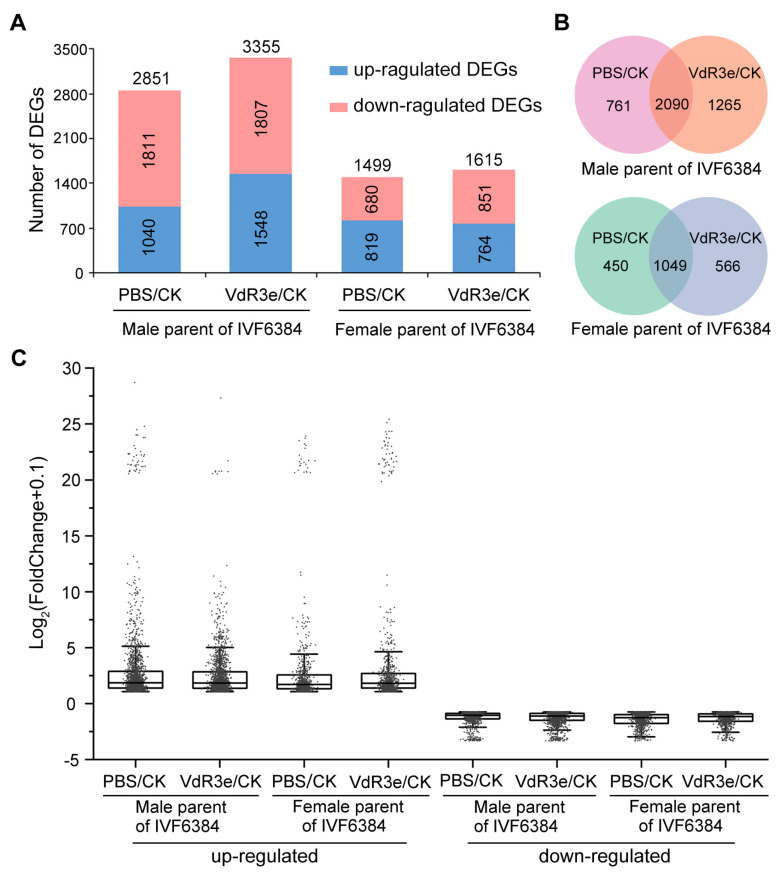
Analysis of differentially expressed genes (DEGs) in IVF6384 parents during VdR3e protein infiltration. (**A**) Statistics on the number of DEGs that are upregulated and downregulated in IVF6384 male and female parents under three conditions: control CK (no treatment), infiltration of PBS buffer, and infiltration of VdR3e protein, with pairwise comparisons. (**B**) The Venn diagram of the DEGs indicated unique and common DEGs in CK vs. PBS and CK vs. VdR3e of IVF6384 male and female parents. (**C**) The fold change in DEGs in CK vs. PBS and CK vs. VdR3e of IVF6384 male and female parents.

**Figure 3 plants-14-01243-f003:**
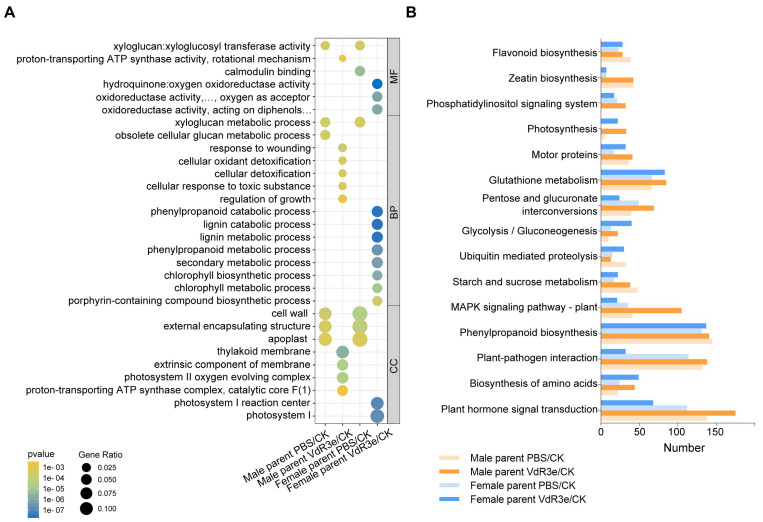
Functional analysis of IVF6384 parents’ response to VdR3e protein infiltration. (**A**) Gene ontology (GO) annotation predicted DEGs in IVF6384 male and female parents, respectively, versus the predicted protein-coding genes from the whole genome. Numbers represented the percentage of enriched items. (**B**) The potential pathways were predicted by the KEGG using DEGs in IVF6384 male and female parents, respectively.

**Figure 4 plants-14-01243-f004:**
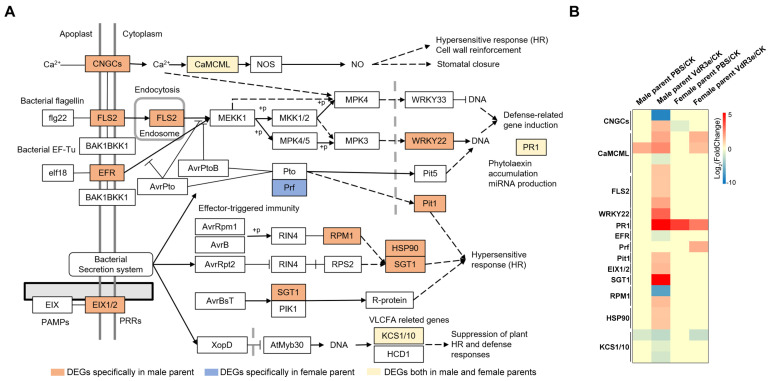
DEGs were assigned in plant–pathogen interaction. (**A**) Demonstration of enrichment of DEGs in plant–pathogen interaction pathway with a *p*-value < 0.05. (**B**) Heatmap of the expression patterns for DEGs induced by VdR3e. Colored squares indicate expression levels of the selected genes from −5 (blue) to 5 (red) normalized by log_2_ (TPM + 1).

**Figure 5 plants-14-01243-f005:**
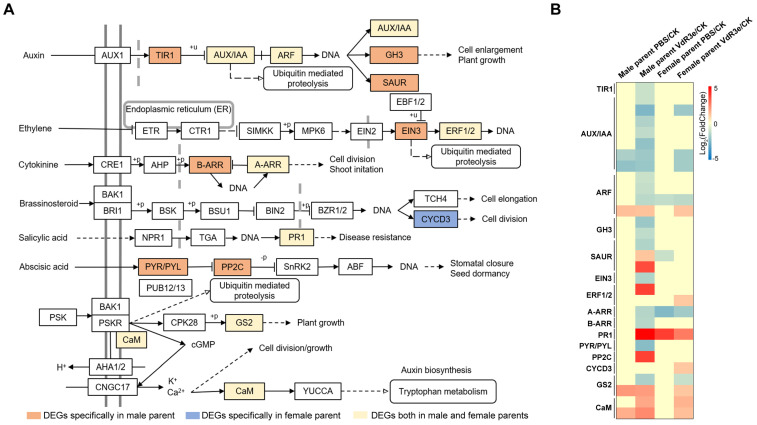
DEGs were assigned in plant hormone signal transduction. (**A**) Demonstration of enrichment of DEGs in the plant hormone signal transduction pathway with a *p*-value < 0.05. (**B**) Heatmap of the expression patterns for DEGs induced by VdR3e. Colored squares indicate expression levels of the selected genes from −10 (blue) to 5 (red) normalized by log_2_ (TPM + 1).

**Figure 6 plants-14-01243-f006:**
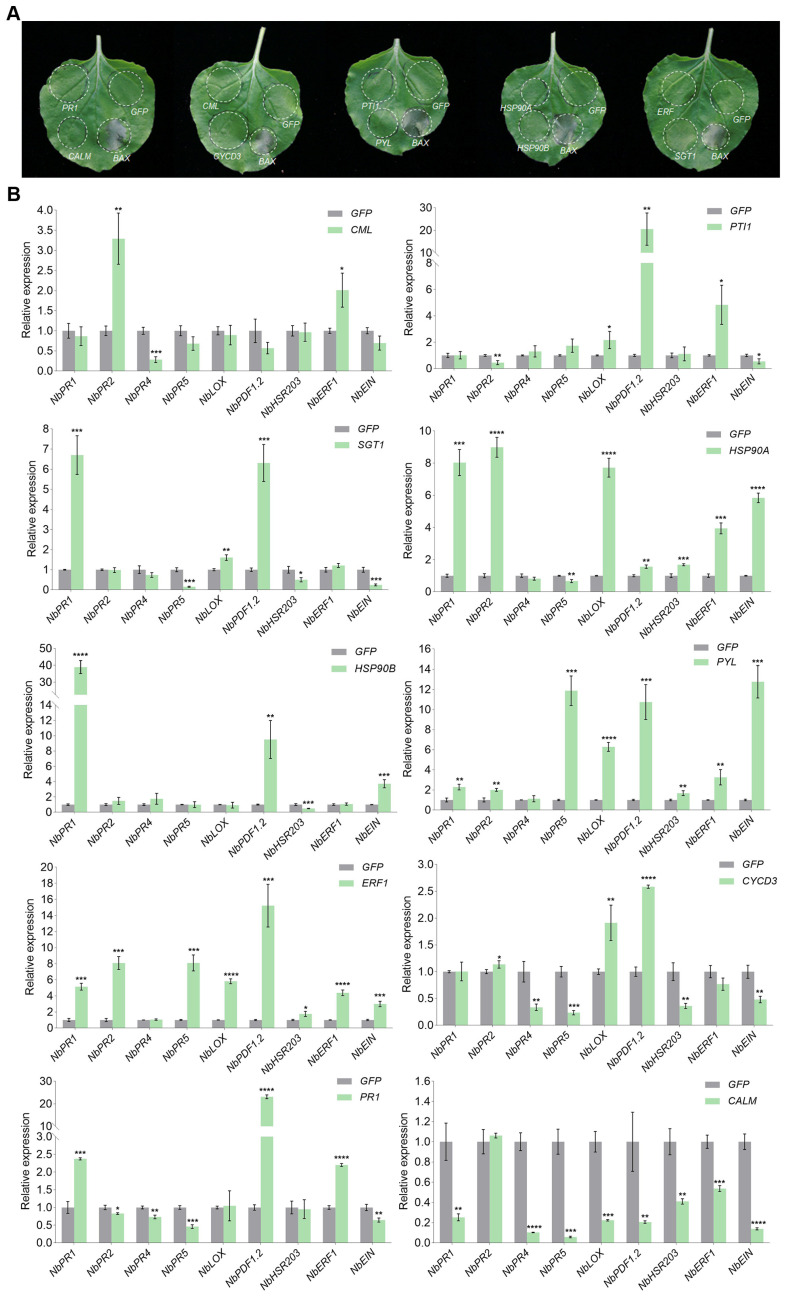
Analysis of immune function of differentially expressed genes specifically induced by VdR3e in IVF6384 parents. (**A**) Necrotic phenotype caused by infiltration of the VdR3e protein into *N. benthamiana* leaves at 5 days post infiltration (dpi). (**B**) qRT-PCR analysis of the expression levels of immune-related genes. Error bars indicate ±SD of three biological replicates, with each measured in triplicate. * indicates significant differences (*p* < 0.05); ** indicates significant differences (*p* < 0.01); *** indicates significant differences (*p* < 0.001); **** indicates significant differences (*p* < 0.0001).

## Data Availability

Data are contained within the article and Appendix A.

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
