# Peer review of "Transcriptomic Profiling Reveals Regulatory Pathways of Tomato in Resistance to Verticillium Wilt Triggered by VdR3e"

_plants, 2025, doi:10.3390/plants14081243_

Round 1
Reviewer 1 Report
Comments and Suggestions for Authors
In the manuscript named “Transcriptomic Profiling Reveals Gene Networks of Tomato in Resistance to Verticillium Wilt Triggered by VdR3e”, authors have comprehensively performed RNA-seq analysis in tomato response to Verticillium dahlia, their results have shown VdR3e with important roles in this process. Their findings would be helpful for tomato genetic breeding works in future. However, there were some comments about it.
(1) Authors have claimed that their data was deposited in PRJNA1205222, but the data could be assessed, please check it.
(2) Authors have missed many descriptions in method section, for example, how many samples in RNA-seq, and how to mapping reads to genome sequences, or other sequences, etc. In addition, authors have described as using Cuffdiff for DEG analysis, but this software was released on 2010, which would be bias in more large reads analysis, for example, Illumina HiSeq X-Ten data.
(3) According to authors description, IVF6384 parents were all tomato lines, they could be compared with same reference genome sequences, but authors just compared control or VdR3e groups, not compared them in different lines, which would be effective for exploring molecular mechanism. Why not?
(4) The title was “Gene Networks”, but manuscript was displayed with some pathways, the “Gene networks” would not be appropriate here.
(5) Authors have described “…resistance-related KEGG pathways enriched by DEGs”, but authors have not provided p-value for this conclusion, similar was figure 4 and figure 5. In addition, authors have merged DEGs from male and female parents, but they didn’t explain their differences or divergence between these two lines. Ps, the merge with more DEGs, but the background genes were also enlarged in statistics analysis.
(6) The results have shown PDF gene with intense reaction, see figure 6B, but authors have overlooked it in discussion section, please check it.
Author Response
Comments 1: Authors have claimed that their data was deposited in PRJNA1205222, but the data could be assessed, please check it.
Response 1: The data we have submitted to NCBI and the BioProject accession number is PRJNA1205222, the release date is 2025-07-01. Thak you for your comment. We have applied for release this data right now.

Comments 2: Authors have missed many descriptions in method section, for example, how many samples in RNA-seq, and how to mapping reads to genome sequences, or other sequences, etc. In addition, authors have described as using Cuffdiff for DEG analysis, but this software was released on 2010, which would be bias in more large reads analysis, for example, Illumina HiSeq X-Ten data.
Response 2: Thank you for your helpful comments. We have added description of “4. Materials and Methods” in revised manuscript. In response to your comments of RNA-seq and DEG analysis, we added “There were 6 groups that male parent infiltrated with 5 μg/mL VdR3e protein, in-filtrated with PBS buffer and no treatment (CK), and female parent infiltrated with 5 μg/mL VdR3e protein, infiltrated with PBS buffer and no treatment (CK). 3 replicates in each group, with a total of 18 RNA samples. The RNA samples were sent to the transcriptome for sequencing, and the returned sequencing data were collated. Clean reads were obtained by using the Illumina HiSeq X-Ten to removing the adapter and low-quality reads (quality score > Q20). Clean reads were mapped to tomato reference genome sequences using Hierarchical Indexing for Spliced Alignment of Transcripts (HISAT) data comparison software [59].” in line 462-470; and rewrote “DEseq2 R package (1.46.0) was used to calculate the FPKM of genes in each sample [60].” in line 476-477.
Comments 3: According to authors description, IVF6384 parents were all tomato lines, they could be compared with same reference genome sequences, but authors just compared control or VdR3e groups, not compared them in different lines, which would be effective for exploring molecular mechanism. Why not?
Response 3: There are many genes differentially expressed in any two different tomato lines during growth and stress response, including IVF6384 parents. Actually, we would like to compare the differences between IVF6384 male parent and female parent defense against VdR3e infiltration in this paper. Thus, we compared genes differentially expressed in IVF6384 parents after infiltration VdR3e with no infiltration (control) to discover the differences between IVF6384 male and female parents. We are sorry that we did not describe clearly in manuscript. We added descriptions of this part in resubmitted draft. See in line 238-242, 256-258, 366-398.
Comments 4: The title was “Gene Networks”, but manuscript was displayed with some pathways, the “Gene networks” would not be appropriate here.
Response 4: Thank you for your comment. We corrected the “Gene Networks” to “Regulatory Pathways” in revised manuscript.
Comments 5: Authors have described “…resistance-related KEGG pathways enriched by DEGs”, but authors have not provided p-value for this conclusion, similar was figure 4 and figure 5. In addition, authors have merged DEGs from male and female parents, but they didn’t explain their differences or divergence between these two lines. Ps, the merge with more DEGs, but the background genes were also enlarged in statistics analysis.
Response 5: Thank you for your suggestion, p-value for conclusion were in the figure note. We are sorry that we have omitted the specific description of the differences between parents, which we added in the revised manuscript. In addition, the DEGs from male and female parents showed in Figure 4 and Figure 5 were identified from male and female parents independently, and then match to KEGG pathways. Thus, the background genes cannot enlarge when they merged. We are also sorry that we didn’t explain the differences or divergence between male and female parents. We added more explanation of these differences in the part of Results and Discussion in resubmitted manuscript. Thank you again for your valuable comments.
Comments 6: The results have shown PDF gene with intense reaction, see figure 6B, but authors have overlooked it in discussion section, please check it.
Response 6: Thank you for valuable suggestion. We added relevant descriptions in the revised manuscript. Please see in line 410-414.

Reviewer 2 Report
Comments and Suggestions for Authors
Dear Authors
I am writing to you in regard to manuscript ID-plants-3537004 entitled: “Transcriptomic Profiling Reveals Gene Networks of Tomato in Resistance to Verticillium Wilt Triggered by VdR3e”
The contents of this paper fully accomplished the scope of Plants MDPI journal and, in my opinion the manuscript could contribute to the general advancing knowledge with novel and interesting findings about the research of tomato resistance to VW.
Nevertheless, in my opinion the manuscript should be not accepted in the current form for the following reasons:
- The authors should provide more recent and pertinent bibliography about the topic (see annotated PDF) above all for the introduction section. In some cases, the bibliography is scarce especially for the general description of tomato diseases and Verticillium fungus. Please see annotated PDF including my suggestions and comments.
- What statistical package used? please, provide if it is possible more details in the statistical section.
- The authors should use the Plants template, which requires figures and tables to be embedded in the main body of the text;
- I suggest to the authors to improve the resolution and/or enlarge of images (especially Figg. 1, 3, 4 and 5).
- I’m not fluent in English but the paper is in many sentences too hard to follow for the reader. I suggest a further proofreading by an English mother-tongue or by a colleague fluent in English.
- For the suggestions/comments see Annotated PDF including also minor suggestions.
Therefore, in my opinion the paper should not be considered for publication in its current form. Only following adequate addressing all comments/requests and through English revision of the text, the paper could be reconsidered for the publication.

Although I’m not fluent in English, I think the paper is in many sentences too difficult to follow for the reader. So, I suggest a further proofreading by an English mother-tongue or by a colleague fluent in English.
Author Response
Comments 1: The authors should provide more recent and pertinent bibliography about the topic (see annotated PDF) above all for the introduction section. In some cases, the bibliography is scarce especially for the general description of tomato diseases and Verticillium fungus. Please see annotated PDF including my suggestions and comments.
Response 1: Thanks to your detailed suggestions. We carefully referred to your notes and added new related references. We also added descriptions of tomato diseases and Verticillium fungus in the Introduction Section. Please see in line 84-87, 116-121.
Comments 2: What statistical package used? please, provide if it is possible more details in the statistical section.
Response 2: We added more details of the statistical analysis in Materials and Methods. We used Parametric student’s t-test was used to determine whether the expression of the marker gene were statistically different from GFP and DEG after transient transformation of tobacco. Please see in line 513-515.
Comments 3: The authors should use the Plants template, which requires figures and tables to be embedded in the main body of the text;
Response 3: Thank you for your helpful comment. We revised the manuscript using the “Plants” template and the figures embedded in the main body of the text.
Comments 4: I suggest to the authors to improve the resolution and/or enlarge of images (especially Figg. 1, 3, 4 and 5).
Response 4: We are sorry that the unsharp image in PDF version automatically generated. We also submitted the figures in the document of “Figures.zip”, and the quality of each figure is fit to publish (each image is larger than 1 Mb).
Comments 5: I’m not fluent in English but the paper is in many sentences too hard to follow for the reader. I suggest a further proofreading by an English mother-tongue or by a colleague fluent in English.
Response 5: Thank you for your suggestion. We sent our manuscript to English editing service, which help us to edit and improve the English.
Comments 6: For the suggestions/comments see Annotated PDF including also minor suggestions.
Response 6: Thank you for suggestions. We corrected all questions raised by your helpful comments.
Comments 7: Therefore, in my opinion the paper should not be considered for publication in its current form. Only following adequate addressing all comments/requests and through English revision of the text, the paper could be reconsidered for the publication.
Although I’m not fluent in English, I think the paper is in many sentences too difficult to follow for the reader. So, I suggest a further proofreading by an English mother-tongue or by a colleague fluent in English.
Response 7: Thank you for your suggestion. We corrected all questions raised by reviewers’ comments and sent our manuscript to English editing service, which help us to improve the English and the quality of revised draft.

Round 2
Reviewer 1 Report
Comments and Suggestions for Authors
Thanks for authors work, most of my comments were well addressed in revision. But the figure 2c, there were a great many extreme values in the figure 2c, if the FPKM values were added 0.1 or 1, the occurrence of extreme values can be reduced. Good luck.
Author Response
Comments 1: Thanks for authors work, most of my comments were well addressed in revision. But the figure 2c, there were a great many extreme values in the figure 2c, if the FPKM values were added 0.1 or 1, the occurrence of extreme values can be reduced. Good luck.
Response 1: Thank you for your helpful suggestion. We modified the Figure 2C to reduce the extreme values which FPKM values added 0.1. The revised Figure 2 has been uploaded in resubmitted version.

Reviewer 2 Report
Comments and Suggestions for Authors
Dear Authors,
Your efforts improved the quality of manuscript and thus the overall merit to make it publishable in the Journal
Author Response
Comments 1: Your efforts improved the quality of manuscript and thus the overall merit to make it publishable in the Journal.
Response 1: Thank you for your valuable comments to improve our manuscript.
